# Effects of Carbonyl Iron Powder (CIP) Content on the Electromagnetic Wave Absorption and Mechanical Properties of CIP/ABS Composites

**DOI:** 10.3390/polym12081694

**Published:** 2020-07-29

**Authors:** Wenwen Lai, Yan Wang, Junkun He

**Affiliations:** School of Materials Science and Engineering, Wuhan Institute of Technology, Wuhan 430074, Hubei, China; lightwee@163.com (W.L.); mrhejk@163.com (J.H.)

**Keywords:** CIP/ABS composites, electromagnetic wave absorption, mechanical properties, functional material, FDM

## Abstract

Three-dimensional (3D) printing technology has proven to be a convenient and effective method to fabricate structural electromagnetic wave (EMW) absorbers with tunable EMW absorption properties. To obtain a functional material with strong EMW absorbing performance and excellent mechanical properties for fused deposition modeling (FDM) 3D printing technology, in this work, carbonyl iron powder (CIP)/acrylonitrile-butadiene-styrene copolymer (ABS) composites with different CIP contents were prepared by the melt-mixing process. The effects of the CIP content on the EMW absorption and mechanical properties of CIP/ABS composites were investigated. The CIP/ABS composite with a CIP content of 40 wt.% presented the lowest reflection loss (RL) of −48.71 dB for the optimal impedance matching. In addition, this composite exhibited optimal mechanical properties due to the good dispersion of the CIPs in the matrix ABS. Not only were the tensile and flexural strength similar to pure ABS, but the tensile and flexural modulus were 32% and 37% higher than those of pure ABS, respectively. With a CIP content of 40 wt.%, the CIP/ABS composite proved to be a novel functional material with excellent EMW absorbing and mechanical properties, providing great potential for the development of structural absorbers via FDM 3D printing technology.

## 1. Introduction

Nowadays, electromagnetic wave (EMW) radiation has attracted much attention due to its great damage to the health of human beings and information safety [1,2,3,4,5,6]. To solve these problems, there is an urgent demand for the development of electromagnetic wave absorbing materials (EMWMs) with high EMW absorbing performance [7,8,9,10,11]. Over the past decade, a number of EMWMs have been developed and reported for these purposes, such as metal magnetic particles, metal oxides, carbon materials, and ceramic materials [12,13,14,15,16]. Recently, polymeric EWAMs, which were fabricated by adding one or more types of absorbing agents to the polymer matrix, have been widely investigated due to their outstanding mechanical properties and processability [17,18,19,20,21,22]. Among these absorbents, carbonyl iron powders (CIPs) have been widely used owing to their low price, high specific saturated magnetization, and strong absorption capacity [23,24,25,26,27]. However, a large additive amount of absorbents have been considered a necessity for polymeric EWAMs with excellent EMW absorbing performance, which contributes to the poor mechanical property and bad processability of EWAMs due to the extremely weak compatibility of the absorbents and the polymer matrix [28]. To this end, much effort has been devoted to improving the EMW-absorbing performance of polymeric EWAMs with low filler concentration. The modification of magnetic fillers, employment of multiple absorbents, and structural design of EWAMs have proven to be effective in reducing the reflection loss (RL) and broadening the effective absorption bandwidth (EAB; RL <−10 dB, for over 90% EMW energy absorbed) [29,30,31,32,33,34,35,36,37,38,39,40,41,42,43]. In particular, structural EWAMs have drawn deep attention not only due to their ability to adjust the impedance matching of absorbers and enhance the absorbing performance but also due to excellent mechanical properties [44,45,46]. Nevertheless, it is difficult and costly to achieve the free design of various structures of structural EWAMs by traditional manufacturing systems.

Recently, three-dimensional (3D) printing technology has been more widely used as compared to traditional manufacturing tools due to its inherent capability of producing unique and complex geometries utilizing computer-aided design (CAD) [47,48,49,50,51]. Among various other 3D printing technologies, such as stereolithography (SLA), direct ink writing (DIW), digital light processing (DLP), and selective laser sintering (SLS), fused deposition modeling (FDM) is considered the most versatile technology due to its high safety, low cost, and convenient fabrication of parts involving complex geometries [52,53,54,55]. In the FDM system, a thermoplastic monofilament is melted inside a heated nozzle and deposited in a layer-by-layer approach onto the build platform according to the predefined CAD model. The outstanding productivity and flexibility of FDM 3D printing technology has significantly developed modern structural materials involving the areas of aerospace, automotive, and medical engineering [56,57,58,59]. Designing various microstructures via FDM technology has proven to be a significantly convenient and effective method to fabricate structural EWAM absorbers with tunable EMW absorption properties [60]. However, the development of such FDM-printed structural EWAM absorbers is limited by the lack of 3D printing functional materials with strong absorption capacity and excellent mechanical properties.

Therefore, this work aims to provide a functional material with strong EMW-absorbing performance and excellent mechanical properties for FDM 3D printing technology. For the FDM process, acrylonitrile-butadiene-styrene copolymer (ABS), a high-strength engineering plastic, has been considered the most widely used material for its superior mechanical properties and good processing fluidity [61,62,63,64]. In addition, the relatively stable two-phase interface formed by acrylonitrile and styrene grafted on polybutadiene contributes to the good compatibility of fillers and ABS, which provides favorable conditions for the blending modification of fillers and ABS [65]. In this work, CIP/ABS composites with varying CIP contents were prepared by the melt-mixing process. The electromagnetic and mechanical properties of CIP/ABS composites were measured using a vector network analyzer and a universal testing machine, respectively. The effects of the CIP content on the EMW-absorbing and mechanical properties of the CIP/ABS composites were investigated. The fracture interface of the CIP/ABS composites was observed using scanning electron microscopy (SEM) micrographs.

## 2. Experimental Setup

### 2.1. Materials

The CIPs (BD-MZ-1) were obtained from Aerospace Science and Industry Wuhan Magnetoelectric Co., Ltd., Wuhan, China. The ABS pellets were purchased from LG Chem Co., Ltd., Seoul, Korea. The basic properties and electromagnetic parameters of the CIPs are presented in Table 1.

### 2.2. Preparation of the CIP/ABS Composites

CIP/ABS composites with different CIP contents were prepared via a melt-mixing process with a miniature mixer (QE-70A, Wuhan Qien Technology Development Co., Ltd., Wuhan, China), which was fitted with counter-rotating roller impellors. The temperatures of the front plate, middle plate, and backplate of the mixer were set to 220, 228 and 225 °C, respectively. Mixing was performed for 10 min at a rotor speed of 30 rpm. The obtained composites were labeled C-10, C-20, C-30, C-40, C-50, and C-60 with the numbers corresponding to their respective CIPs weight fractions, as shown in Table 2. The CIP/ABS composites were then shredded into small pellets using a mechanical chopper (NPCY-30, Dongguan Najin Machinery Co., Ltd., Dongguan, China) and dried in a vacuum oven at the temperature of 78 °C.

### 2.3. Injection Molding of the CIP/ABS Composites

A microinjection molding machine (M-1200, Wuhan Qien Technology Development Co., Ltd., Wuhan, China) was used for the injection molding of the CIP/ABS composites. The barrel temperature of the injection molding machine was set to 235 °C, and the injecting time, mold closing time, and cooling time were set to 5 s, 6 s, and 4 s, respectively. Straight samples with a dimension of 80 mm × 10 mm × 4 mm and dumbbell-shaped specimens with a dimension of 75 mm × 10 mm × 2 mm (gauge length: 25 mm) were fabricated using the different molds for the mechanical flexural and tensile properties tests.

### 2.4. Compression Molding of the CIP/ABS Composites

To obtain the required sample for the measurement of the electromagnetic parameters, compression molding was performed via a hot press (R-3221, Wuhan Qien Technology Development Co., Ltd., Wuhan, China). CIP/ABS composite pellets with a theoretical forming mass content of 110% were placed into the mold and pressed for 15 min at a temperature of 230 °C under a pressure of 10 MPa. The mold was then cooled to generate a concentric annular-shape sample with an inner diameter of 3.04 mm, an outer diameter of 7 mm, and a thickness of 2 mm.

### 2.5. Characterization and Testing

The morphology of the CIP/ABS composite samples was observed using scanning electron microscopy (SEM, JSM-5510 LV, Tokyo, Japan) under an accelerating voltage of 20 kV. The samples were cryo-fractured in liquid nitrogen, and the fractured surfaces were coated with a layer of gold in a vacuum chamber prior to the visualization by SEM. The electromagnetic parameters, complex relative permittivity ($\varepsilon_{r}=\varepsilon \prime -j\varepsilon \prime \prime$), and permeability ($\mu_{r}=\mu \prime -j\mu \prime \prime$) of the pure ABS, C-10, C-20, C-30, C-40, C-50, and C-60 samples were measured by a vector network analyzer (N5224A, Keysight Technologies, Santa Rosa, CA, USA) within the range of 2–18 GHz. The mechanical tensile properties of all the samples were evaluated via a universal testing machine (5960, INSTRON, Boston, MA, USA) at a strain rate of 20 mm/min according to the GBT1040.1-2006 standard. The mechanical flexural property tests were performed at room temperature by a universal testing machine (5960, INSTRON, Boston, MA, USA) at a test speed of 2 mm/min following the GBT 9341-2008 standard.

## 3. Results and Discussion

### 3.1. Electromagnetic Property Analyses

As is well-known to us all, that the EMW absorption properties of materials rely highly on the complex relative permittivity ($\varepsilon_{r}=\varepsilon \prime -j\varepsilon \prime \prime )$ and permeability $\left(\mu_{r}=\mu \prime -j\mu \prime \prime \right)$ [66]. Herein, ε′ and *μ*′ represent the storage capability of electric and magnetic energy, respectively, and ε″ and *μ*″ determine the deterioration capability of electric and magnetic energy, respectively [67]. Figure 1a–d shows the frequency dependence of ε′, ε″, *μ*′, and *μ*″ of pure ABS, C-10, C-20, C-30, C-40, C-50, and C-60 in the frequency range of 2–18 GHz. As depicted in Figure 1a–d, it can be clearly seen that an increase in the CIP contents produced higher values of ε′, ε″, *μ*′, and *μ*″ in the frequency range of 2–18 GHz. From Figure 1a, sample C-60 with the highest CIP content presented the largest ε′ of 4.87, showing no evident dependence on the frequency. In comparison, its ε″ values fluctuated from 0 to 0.91 with variations in the frequency, as shown in Figure 1b. In Figure 1c, the *μ*′ values of the samples gradually decreased with increasing frequency, indicating a typical phenomenon induced by the domain-wall motion and relaxation [68]. Figure 1d shows that *μ*″ varied directly with the frequency, suggesting a better deterioration capability of magnetic energy at the high-frequency region. The dielectric loss tangent $\left({\tan\delta }_{E}=\varepsilon \prime \prime /\varepsilon \prime \right)$ and magnetic loss tangent $({\tan\delta }_{M}=\mu \prime \prime /\mu \prime$) are commonly used to describe electromagnetic loss capacity. For an ideal EMW absorber, ${\tan\delta }_{E}$ and ${\tan\delta }_{M}$ are regarded to be immensely large [69]. As can be clearly seen in Figure 1e,f, sample C-60 presented the maximum ${\tan\delta }_{E}$ value of 0.19 and ${\tan\delta }_{M}$ value of 0.47, indicating an enormous potential to attenuate electromagnetic wave energy.

### 3.2. Electromagnetic Wave Absorbing Performance Analyses

According to the transmission-line theory [70], the RL of single-layer absorbers can be calculated using the following equations:(1)$$RL=20lg\left|\frac{Z_{\mathrm{in}}-Z_{0}}{Z_{\mathrm{in}}+Z_{0}}\right|$$
(2)$$Z_{\mathrm{in}}=Z_{0}\sqrt{\frac{\mu_{r}}{\varepsilon_{r}}}\tanh\left(\frac{j2\pi fd}{c}\sqrt{\mu_{r}\varepsilon_{r}}\right),$$
where $Z_{0}$ is the impedance of free space, $Z_{\mathrm{in}}$ is the input characteristic impedance, *f* is the frequency, *c* is the velocity of light, and *d* is the thickness of absorbers. As a result, Figure 2 shows the frequency dependence of the RL of samples C-10, C-20, C-30, C-40, C-50, and C-60 when the thickness increased from 0.5 to 8 mm in the frequency range of 2–18 GHz. As shown in Figure 2a or Figure 2d, sample C-10 presented a minimum RL (RL-min) of −7.13 dB at a corresponding thickness of 8.0 mm, covering a maximum EAB (EAB-max) of 0 GHz, which indicated its weak capability of attenuating electromagnetic waves. As compared to C-10, C-20 with a thickness of 8.0 mm had a significantly enhanced EMW-absorbing performance of a lower RL-min of -18.37 dB and a larger EAB-max of 1.8 GHz (Figure 2b or Figure 2e). As for C-30 in Figure 2c or Figure 2f, a larger EAB-max of 2.6 GHz was obtained at a corresponding thickness of 7.0 mm, which covered a part of the Ku-band (12–18 GHz). It can be clearly seen in Figure 2g or Figure 2j that sample C-40 presented an EAB-max of 3.3 GHz at a corresponding thickness of 7.0 mm. Interestingly, it also acquired the lowest RL-min of −48.71 dB among all the samples, suggesting its excellent EMW absorption performance. In Figure 2h or Figure 2k, C-50 presented an RL-min of −28.99 dB and higher EAB-max of 4.1 GHz at a thickness of 8.0 mm, also covering a part of the Ku-band. With the highest content of CIPs, C-60 showed the largest EAB-max of 4.6 GHz at a thickness of 8.0 mm, covering a part of the X-band (8–12 GHz) and Ku-band. These results are summarized in Table 3.

According to Figure 2 and Table 3, the EMW absorption properties of the CIP/ABS composites gradually enhanced with an increase in the CIP content possibly due to proper impedance matching and strong attenuation. As is well-known, the impedance matching and attenuation constant are regarded as the two most critical factors for the EMW absorption properties of absorbers [71]. As for the ideal condition of impedance matching, which determines the capability of incident waves entering into an absorber, |$Z_{in}$| or |$Z_{in}-1$| was expected to be closer to 1 or 0 [72]. The |$Z_{in}-1$| values of the samples at the optimal thickness corresponding to RL-min were calculated according to equation (2), as shown in Figure 3a. It can be clearly seen that |$Z_{in}-1$| of sample C-40 was closest to 0 among all the samples, indicating that it presented the optimal impedance matching, which contributed to the lowest RL-min of −48.71 dB among all the samples. In addition, taken as a whole in the frequency range of 2–18 GHz, the |$Z_{in}-1$| values of the samples moved closer to 0 as the CIP content increased, indicating the gradually enhancing impedance matching.

The dissipation capability of the electromagnetic waves entering into the material was evaluated by the attenuation constant, α, which can be expressed as follows [73,74,75]:(3)$$\alpha =\frac{\sqrt{2}\pi f}{c}\sqrt{\left(\mu \prime \prime \varepsilon \prime \prime -\mu^{\prime }\varepsilon^{\prime }\right)+\sqrt{{\left(\mu \prime \prime \varepsilon \prime \prime -\mu \prime \varepsilon \prime \right)}^{2}+{\left(\mu \prime \varepsilon \prime \prime -\mu \prime \prime \varepsilon \prime \right)}^{2}}}.$$

As depicted in Figure 3b, the attenuation constant α of the samples gradually increased with increasing CIP content, indicating the gradually enhancing dissipation capacity of the EMW energy. Based on the highest content of the CIPs, the attenuation constant α of sample C-60 was much larger than other samples and thus presented the largest EAB-max of 4.6 GHz among all the samples.

In addition, Figure 2 indicates that the RL attenuation peaks of samples shifted to lower frequencies with increasing sample thickness. These phenomena may be explained by quarter-wavelength (λ/4) cancellation model, which describes the relationship between the thickness of absorber, $d_{m}$, and the corresponding frequency at the RL absorption peak, $f_{m}$, and can be defined by the following equation [76]:(4)$$d_{m}=\frac{nc}{4f_{m}\sqrt{\left|\mu_{r}\varepsilon_{r}\right|}}\;(n=1,\;3,\;5,\;\ldots ).$$

Figure 4 presents the theoretical *λ*/4 cancellation model of sample C-60. The given absorber thickness of C-60 versus the corresponding frequency at the RL absorption peak in Figure 2l is denoted as blue dots. Obviously, these absorber thicknesses fitted well with the λ/4 curve.

### 3.3. Mechanical Properties of the CIP/ABS Composites

The mechanical tensile and flexural properties of the CIP/ABS composites are presented in Figure 5. Figure 5a shows that the tensile strength of CIP/ABS composites first increased and then decreased with an increase in the CIP content from 10 wt.% to 60 wt.%. When 10 wt.% CIP was added, the CIP/ABS composite presented a tensile strength of 40.94 MPa, which was approximately 6% lower than that of pure ABS (43.56 MPa). When the CIP content increased to 40 wt.%, the tensile strength of the composite reached the maximum value of 44.38 MPa and was slightly higher than that of pure ABS. With the highest CIP content of 60 wt.%, the tensile strength of the composite decreased to 36.56 MPa, which was 16% lower than that of pure ABS. In addition, similar relationships were observed between the CIP content versus flexural strength and the CIP content versus tensile strength. As shown in Figure 5c, the CIP/ABS composite with a CIP content of 40 wt.% also presented the largest flexural strength of 61.13 MPa, which was slightly higher than that of pure ABS (60.34 MPa). When the CIP content increased to 60 wt.%, the flexural strength of the composite also decreased to 50.58 MPa, which was 16% lower than that of pure ABS. The effects of the CIP content on tensile and flexural modulus are shown in Figure 5b or Figure 5d. Both the tensile and flexural moduli increased with the CIP content. The CIP/ABS composite with a CIP content of 40 wt.% presented a tensile modulus of 1701 MPa and a flexural modulus of 2745 MPa, which were 32% and 37% higher than those of pure ABS, respectively. When the highest CIP content of 60 wt.% was added, both the tensile and flexural moduli of the composite reached the maximum values and were 83% and 79% higher than those of pure ABS, respectively.

Sample C-40 with a CIP content of 40 wt.% presented similar tensile and flexural strengths to pure ABS, as well as 32% and 37% higher tensile and flexural moduli as compared to those of pure ABS, respectively. All of these results indicate the CIP/ABS composite with a CIP content of 40 wt.% presented the optimal mechanical property possibly due to the excellent dispersion of the CIPs in the ABS matrix. According to Figure 6, the spherical-shaped CIPs were uniformly dispersed in the matrix ABS. On the one hand, the CIP particles acted as physical crosslinking points in the material, inhibiting the movement of molecular chains and thus enhancing its mechanical properties [77]. On the other hand, the CIP particles in the material were capable of pinning the generated cracks on the material when the material was subjected to an external load. The direction of cracks propagation was changed due to the pinning effect, thus generating many small cracks from larger cracks, which were able to absorb more external energy and subsequently enhance the mechanical properties [78,79,80,81]. The pinning effect could not be achieved when the CIP content was less than 40 wt.%, such that slightly reduced mechanical properties were obtained. In addition, CIP particle agglomeration produced many defects inside the material when the CIP content was more than 40 wt.%, which also reduced the mechanical properties.

## 4. Conclusions

The EMW-absorption properties of the CIP/ABS composites were gradually enhanced with increasing CIP contents. CIP/ABS composites with a CIP content of 60 wt.% presented the largest EAB (EAB; RL < −10 dB, for over 90% EMW energy absorbed) of 4.6 GHz at a corresponding thickness of 8.0 mm, which was attributed to its large attenuation constant, α. Based on the optimal impedance matching, the CIP/ABS composites with a CIP content of 40 wt.% presented the lowest RL of −48.71 dB at a corresponding thickness of 7.5 mm. In addition, the composites with a CIP content of 40 wt.% presented the optimal mechanical property due to optimal CIP dispersion in the ABS matrix. On the one hand, the CIP particles acted as physical crosslinking points in the material, inhibiting the movement of molecular chains. On the other hand, the CIP particles in the material were capable of pinning the generated cracks on the material when the material was subjected to the external load. Not only were the tensile and flexural strength similar to that of pure ABS, but the tensile and flexural modulus were 32% and 37% higher than those of pure ABS, respectively. This work demonstrated that the CIP/ABS composite with a CIP content of 40 wt.% was able to achieve strong EMW-absorbing performance and excellent mechanical property requirements, showing a promising application prospect for the development of structural absorbers via FDM 3D printing technology.

## Figures and Tables

**Figure 1 polymers-12-01694-f001:**
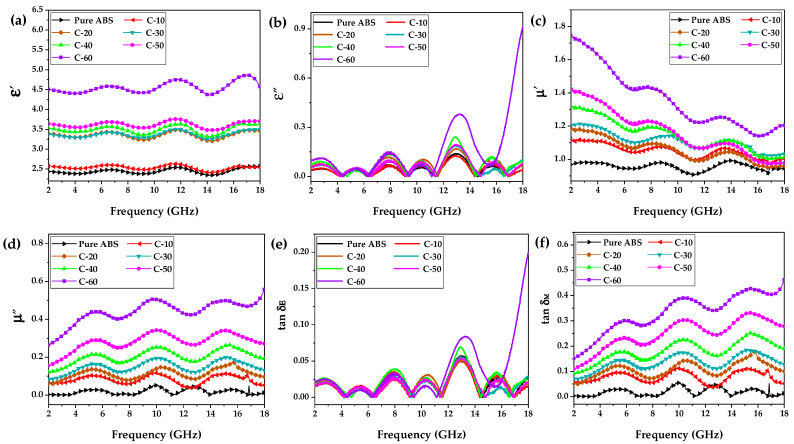
Electromagnetic properties of pure ABS, C-10, C-20, C-30, C-40, C-50, and C-60. (**a**) Real parts (ε′) and (**b**) imaginary parts (ε″) of the complex relative permittivity. (**c**) Real parts (μ′) and (**d**) imaginary parts (μ″) of the complex relative permeability. (**e**) The dielectric loss tangent ${\tan\delta }_{E}$. (**f**) The magnetic loss tangent ${\tan\delta }_{M}.$

**Figure 2 polymers-12-01694-f002:**
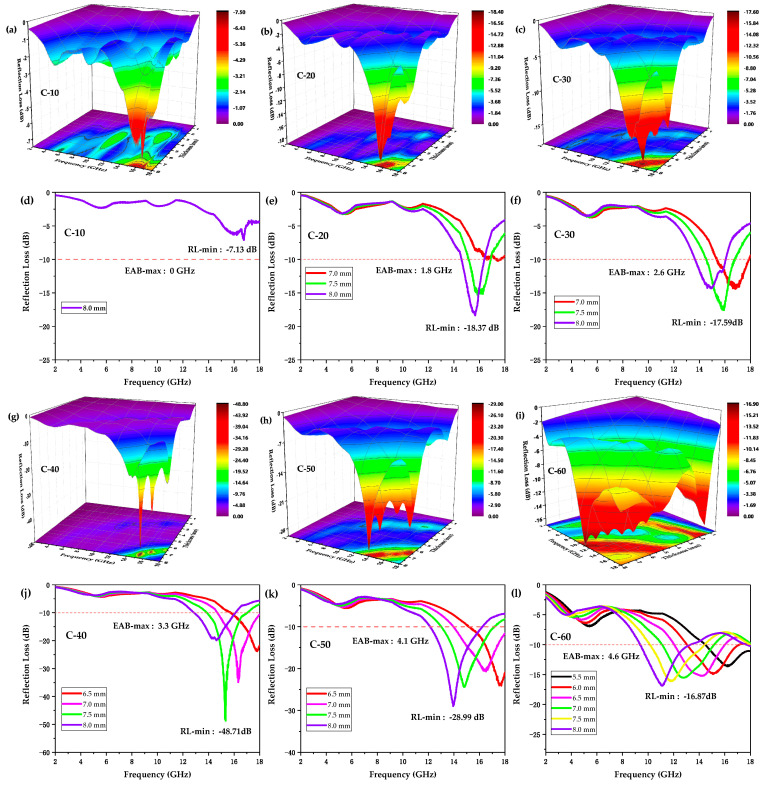
3D theoretically calculated reflection loss (RL) spectra of CIP/ABS composites with varying CIP contents when the thickness increased from 0.5 mm to 8 mm, and the corresponding 2D RL plots. (**a**,**d**): sample C-10. (**b**,**e**): sample C-20. (**c**,**f**): sample C-30. (**g**,**j**): sample C-40. (**h**,**k**): sample C-50. (**i**) and **(l):** sample C-60.

**Figure 3 polymers-12-01694-f003:**
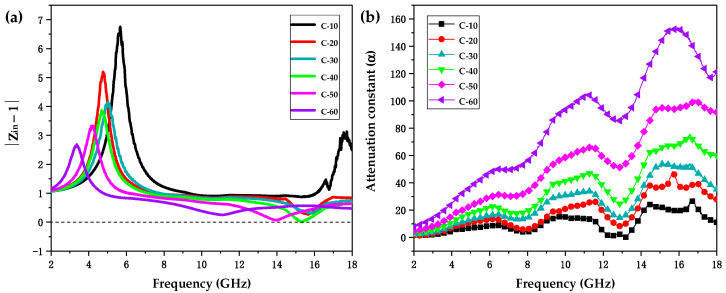
(**a**) The |$Z_{in}-1$| values of samples C-10, C-20, C-30, C-40, C-50, and C-60 at the optimal thickness corresponding to RL-min in the frequency range of 2–18 GHz. (**b**) The attenuation constant α of samples C-10, C-20, C-30, C-40, C-50, and C-60 in the frequency range of 2–18 GHz.

**Figure 4 polymers-12-01694-f004:**
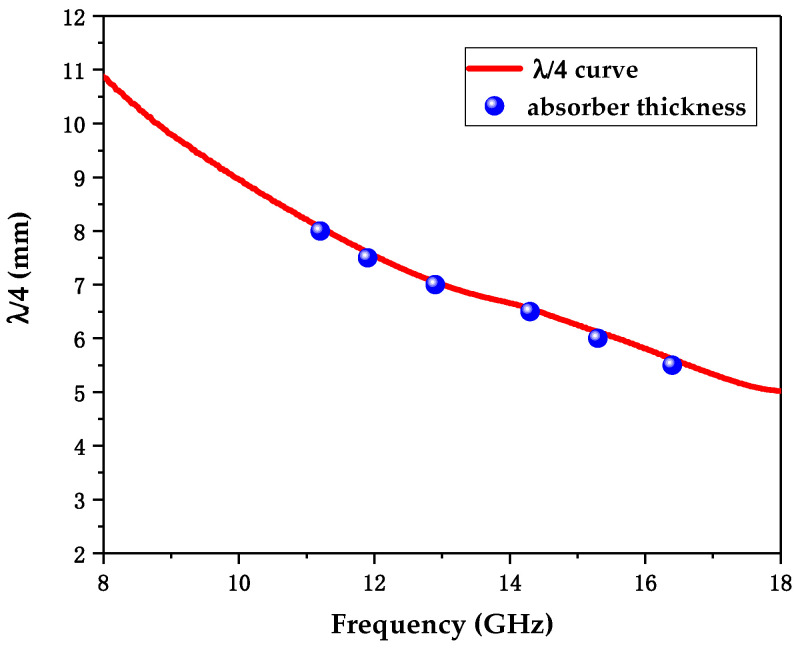
The *λ*/4 cancellation model of sample C-60 and the absorber thickness of C-60 versus the corresponding frequency at the RL absorption peak (denoted as blue dots).

**Figure 5 polymers-12-01694-f005:**
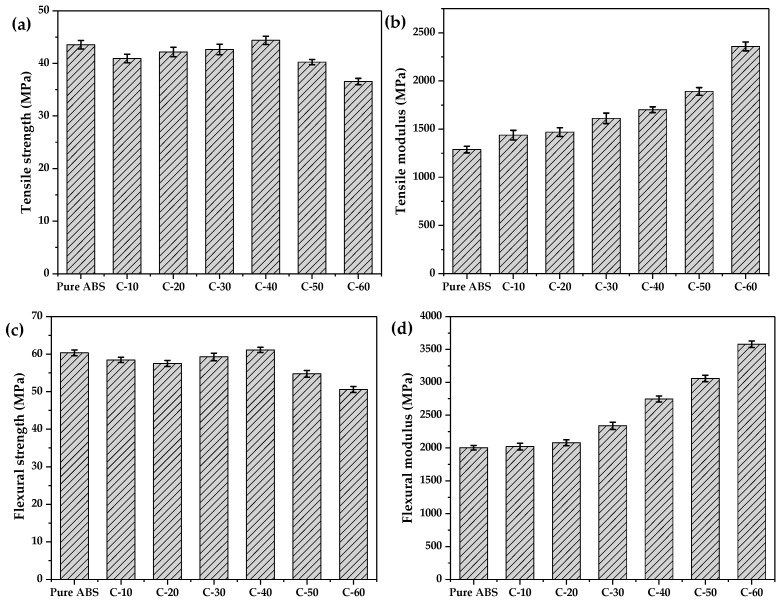
Mechanical properties of CIP/ABS composites with different CIP content. (**a**) Tensile strength. (**b**) Tensile modulus. (**c**) Flexural strength. (**d**) Flexural modulus.

**Figure 6 polymers-12-01694-f006:**
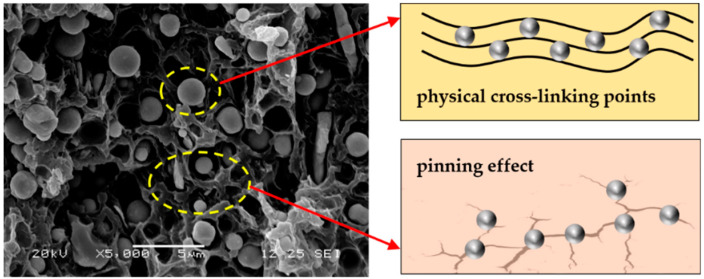
The SEM image of sample C-40 and a hypothetical microscopic mechanism of CIPs on the enhanced mechanical properties.

**Table 1 polymers-12-01694-t001:** Basic properties and electromagnetic parameters of BD-MZ-1 carbonyl iron powders (CIPs).

Particle Sizes (μm)	Tap Density $\left(g/cm^{3}\right)$	Electromagnetic Parameters (Part of Frequencies)				
		Frequency (GHz)	ɛ′	ɛ″	*μ*′	*μ*″
<10	2.6–3.1	2	20.75	0.43	5.10	2.24
		8	19.46	0.18	2.12	2.59
		18	20.88	0.66	0.65	1.52

**Table 2 polymers-12-01694-t002:** Description of CIP/ acrylonitrile-butadiene-styrene copolymer (ABS) composites.

Samples	ABS (wt.%)	CIPs (wt.%)
Pure ABS	100	0
C-10	90	10
C-20	80	20
C-30	70	30
C-40	60	40
C-50	50	50
C-60	40	60

**Table 3 polymers-12-01694-t003:** EMW absorption properties of samples C-10, C-20, C-30, C-40, C-50, and C-60. EAB: effective absorption bandwidth.

Samples	RL-Min (dB)	EAB-Max (RL<−10 dB, GHz)
C-10	−7.13	0
C-20	−18.37	1.8
C-30	−17.59	2.6
C-40	−48.71	3.3
C-50	−28.99	4.1
C-60	−16.87	4.6

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
