# Peer review of "Effects of Carbonyl Iron Powder (CIP) Content on the Electromagnetic Wave Absorption and Mechanical Properties of CIP/ABS Composites"

_polymers, 2020, doi:10.3390/polym12081694_

Round 1

Reviewer 1 Report

This is an interesting article on studying the EMW absorption properties of the CIP/ABS composites that were gradually enhanced with increasing CIP contents. Following minor corrections are needed:

-The font size of figure 1 should be improved.

-The mechanical properties should be discussed in more detail

-Some of the relevant articles on 3D printing should be cited such as Self-Healing Mechanisms for 3D-Printed Polymeric Structures: From Lab to Reality, Polymers 202012(7), 1534; Materials Today Chemistry 16, 100248 (2020); Applied Materials Today 18, 100490  (2020)

  Author Response

Reviewer 2 Report

The research article presented by the authors is of greater importance in the field of science and technology. The article can be accepted for publication.

The authors need to improve the introduction and emphasize more on 3D printing technology and its uses. The results and discussion needs more elaboration on the effect of the properties with respect to composition of the materails. The concliusion part is fine.

Overall the research article can be accepted for publication.

Reviewer 3 Report

This manuscript described the preparation and characterization of EM wave absorption materials. The authors used CIP, which is commercially applicable, as a magnetic absorption material and ABS as a matrix material. EM wave absorption properties were thoroughly investigated based on coaxial system. 

Overall, this manuscript is well written and interesting. It is acceptable in present form. The only thing that the authors have to revise is adding the information on the gauge length of the samples for tensile strength measurement. 
